# Impact of algorithm choice when using off-the-shelf AI segmentation methods in large-scale medical analyses

Emelie Bäcklin[*1,2,3] and Eva Breznik[2]

[1]Department of Clinical Science, Intervention and Technology, Karolinska Institute, Stockholm, Sweden
[2]Biomedical Engineering and Health Systems, KTH Royal Institute of Technology, Stockholm, Sweden
[3]Department of Biomedical Engineering, Karolinska University Hospital, Stockholm, Sweden

## Abstract

Manual segmentation of medical images is time-consuming and impractical for large-scale studies. This study evaluated the impact of five deep learning–based lung lobe segmentation methods on lung volume and low attenuation volume below -950 Hounsfield units (LAV950) using chest CT scans from 2,579 participants in the SCAPIS cohort. Results showed minimal differences in mean lung volume and LAV950 across methods. AUC-ROC values for detecting chronic airflow limitation (CAL) were consistent for inspiratory CT but varied more for expiratory scans, indicating greater model sensitivity. Overall, segmentation choice had limited influence on downstream analysis of LAV950-derived lung function measures.

## 1 Introduction

Manual segmentation of medical images is a time-consuming task that becomes impractical when thousands of segmentations are required (as e.g. in downstream statistical analyses, where adequate statistical power is desired). Currently, the most effective automatic segmentation methods are based on deep learning (DL). To ensure that a method is suitable for a specific dataset, local training or fine-tuning can be performed.

In medical research, local ground-truth data are often unavailable, and DL models are frequently applied off the shelf without fine-tuning.

In lung and lung lobe segmentation, state-of-the-art DL methods achieve high performance according to the Dice Similarity Coefficient (DSC) and related metrics. The publicly available models are often presumed to be sufficiently generalisable for off-the-shelf application. However, their accuracy in the context of downstream tasks remains uncertain.

We previously demonstrated, in a smaller cohort, that low attenuation volume below -950 Hounsfield units (LAV950) in expiratory chest CT is associated to decline in lung function [1]. The LAV950 reflects the amount of tissue with abnormally low density which is known to indicate lung tissue destruction

---

*Corresponding Author.

in lung emphysema [2].

In this study, we repeat the analysis on a subset of the population-based Swedish CArdioPulmonary bioImage Study (SCAPIS) [3], to evaluate how segmentation variability affects estimation of lung volume, LAV950, and detection of lung function decline with LAV950.

## 2 Methods

### 2.1 Dataset

THe entire SCAPIS [3] comprises a total of 30,154 participants, 50-64 years old at inclusion. From the 5,038 individuals in the Stockholm subcohort, we included all participants that had undergone inspiratory and expiratory CT of the chest as well as dynamic spirometry for lung function assesment. Our cohort included scans and spriometry for 1,328 men and 1,251 women (at mean age 57.5).

### 2.2 Segmentation methods

The following DL methods for lung lobe segmentation were compared. *nnU-Net (UN)* [4]: trained in-house with 59 manually segmented inhale-exhale CT pairs from the SCAPIS Stockholm cohort (this training set was excluded from all analyses). The same locally trained nnU-Net but *with tailored postprocessing (UNP)*: the postprocessing included suppressing or reassigning all non-maximal connected components (CC) to neighboring main lobe CCs. *Lungmask (LM)* [5]: a U-Net–based lung lobe segmentation model trained on 231 cases, including both routine and pathological inspiratory and expiratory CT scans. *TotalSegmentator (TS)* [6]: developed from the nnU-Net framework. Trained on a wide range of anatomical structures, including the lungs, using CT data from 1,228 subjects. *TriSwinUNETR (SP)* [7]: combines three SwinUNETR networks (containing SWIN transformer and U-Net), trained on 6,501 inhale-exhale CT pairs from COPDGene dataset and finetuned on 11 cases with nodules.

### 2.3 Analyses

Average lung volume (in mL) and LAV950 (in %) per lobe and segmentation method was calculated, using all data. With no ground truth available, the

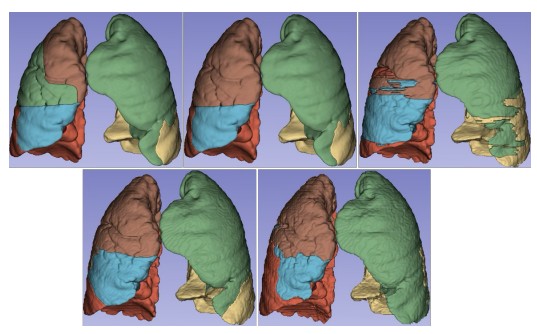

**Figure 1.** Example segmentations: UN, UNP, LM (above), TS and SP (below) respectively.

**Table 1.** Mean lobe volume (mL) per segmentation method.

| Lobe | SP | TS | LM | UN | UNP |
|------|------|------|------|------|------|
| LU | 999.5 | 977.1 | 976.8 | 1021.7 | 1023.6 |
| LL | 914.2 | 902.5 | 897.4 | 923.1 | 922.6 |
| RU | 830.1 | 812.9 | 808.4 | 822.1 | 820.4 |
| RM | 378.3 | 377.0 | 385.9 | 422.7 | 422.4 |
| RL | 1027.0 | 995.3 | 992.2 | 1019.7 | 1020.3 |

per-subject deviation of each method in volume and LAV950 was instead calculated from the mean over all five segmentation methods.

Spirometry data was used to identify subjects with a decline in lung function, defined as chronic airflow limitation (CAL) by $FEV_1/FVC < 0.7$. In our past work, we demonstrated a possibility to discriminate CAL using expiratory LAV950 [1]. Accordingly, the area under the receiver operating characteristic curve (AUC-ROC) for predicting CAL from LAV950 was used to evaluate how segmentation influences downstream analyses. The five lung lobes are denoted by LU (left upper), LL (left lower), RU (right upper), RM (right middle), and RL (right lower).

## 3 Results

Visually, the five methods produced significantly different results (example case in Fig. 1). Despite that, mean lung lobe volumes were similar across methods (Table 1). The maximum deviation between the mean volume and the mean volume of all methods was 3–5% for all lobes except the RM lobe, where it reached 12%. This pattern was also reflected in the mean deviation across methods. The three largest volume deviations from the mean across methods for each lobe were: 443, 425, 372 (LU); 527, 340, 300 (LL); 538, 360, 314 (RU); 334, 266, 243 (RM); 519, 457, 437 (RL). Analysis of LAV950 showed similar patterns. AUC-ROC values for discriminating CAL using LAV950 were comparable across methods and lobes for inspiratory scans. For expiratory scans, LU and RM exhibited similar values across all methods. But in LL, RU, and RL, results were

**Table 2.** AUC-ROC for expiratory LAV950 and CAL, per lobe and segmentation method.

| Lobe | SP | TS | LM | UN | UNP |
|------|------|------|------|------|------|
| LU | 0.74 | 0.74 | 0.75 | 0.74 | 0.74 |
| LL | 0.78 | 0.76 | 0.78 | 0.73 | 0.73 |
| RU | 0.73 | 0.73 | 0.73 | 0.67 | 0.67 |
| RM | 0.69 | 0.69 | 0.70 | 0.68 | 0.68 |
| RL | 0.74 | 0.74 | 0.76 | 0.68 | 0.68 |

comparable across SP, TS, and LM, while UN and UNP showed a decrease (Table 2).

## 4 Discussion

The results indicate that the choice of lung lobe segmentation method has minimal impact on mean lung volume and LAV950. However, given that the individual differences can be large (over 400mL) the impact may be larger for subcohort analyses. The larger differences observed for the RM lobe are expected, as its anatomy is more variable and thus more challenging to segment, even manually.

The AUC-ROC values for identifying CAL with LAV950 were consistent across methods and lobes in inspiratory CT, but showed greater variation in expiratory scans, suggesting that model choice is more critical for those. The in-house post-processing step, taking about 1 minute per image, appears unnecessary for our downstream analyses. Experiments on subcohort level would be needed to determine at what sample size (if at all) the choice of segmentation method substantially influences the analyses.

While our analyses were relatively unaffected, segmentation methods clearly differ in meaningful ways that should be considered in downstream analyses.

## 5 Conclusion

We show that despite noticeable visual differences, the choice of lung lobe segmentation method has little effect on the estimation of lobe volume and LAV950. The modest influence observed on CAL detection in expiratory scans implies that segmentation model selection may be more critical for expiratory analyses. Future work will explore cohort subsets to identify conditions under which segmentation choice may have a more pronounced impact on downstream analyses.

## Acknowledgments

E. Breznik was funded by Hjärt Lung Fonden, Sweden, grant no. 2022/0492.

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
