# OpenReview forum: "Impact of algorithm choice when using off-the-shelf AI segmentation methods in large-scale medical analyses"
_NLDL.org/2026/Abstracts_Track — NLDL 2026 Abstracts_

### Official Review · Reviewer_3fmJ · 2025-10-24

**Soundness:** 2
**Correctness:** 2
**Rating:** 4
**Confidence:** 3

**Summary:**

The abstract looks into how summary measures - denoted as a downstream task - are affected by different choices of pretrained segmentation backbones in lung segmentation. The authors show that the estimated mean lobe volume as predicted by one method deviates from the mean of the remaining methods up to 12% for the RM lobe, and between 3-5% for the remaining lobes. The authors argue that, despite significant visual differences between the segmentations from different models, the resulting downstream task appears relatively unaffected by the choice of backbone.

**Strengths:**

The paper explores interesting points related to the application of deep-learning models in the setting of large-scale medical studies. They analyze the performance across a sufficiently wide range of pretrained models, showing little performance variation in terms of estimated volumes.

They provide a reasonably clear explanation of motivation and an introduction to the pretrained models.

The idea to research the impact of using pretrained models rather than human annotators in this setting is the most interesting and warrants further study; the methodology just needs sharpening.

**Weaknesses:**

The initial claim of the extended abstract is that manual segmentation is not ideal in large-scale medical studies, due to time and economic constraints. The authors seek to evaluate if this can be automated by using pretrained neural networks.  This is a very interesting line of research; however, their methodology is inherently flawed. Large-scale medical research is often exploratory and attempts to find significant, and more reminiscent of their previous publication, where they establish the relation between CAL and volume/LAV950. Thus, if the goal of the research is to estimate the validity of using pretrained models rather than human-annotated labels the results should be focused on the sensitivity of previous conclusions, given that the volume is predicted by a model and not human-annotated.

The authors claim that the observed model deviation for predicting lobe volumes is minimal. However, they show what I would guess is a significant performance difference in terms of predicting CAL from LAV950 - performance varies up to 6 percentage points for RU.

It would be prudent to include variance in Table 1, which would measure both model and anatomical variance, which would be valuable when trying to explain the performance differences in Table 2.

From the extended abstract, it is unclear how one gets from volume and LAV950 to CAL, and it required reading their previous publication to even get an indication of the overall method (discrimination was based on a threshold and not some other ML/DL-based method). It would significantly improve clarity if this were stated.

When defining CAL, the authors introduce acronyms FEV and FVC not previously introduced and is not common knowledge in the ML community.

The authors state noticeable visual differences. The statement is ambiguous, as a reader of the abstract cannot deduce whether they refer to drastically different shapes or textual differences along the segmentation boundary. If it is the latter, it also seems superfluous in this context.

---

### Official Review · Reviewer_qqr9 · 2025-11-02

**Soundness:** 3
**Correctness:** 2
**Rating:** 2
**Confidence:** 2

**Summary:**

The authors look at a large cohort (30k) of inspiratory and expiratory lung CT scans. They use a variety of off-the-shelf networks and finetuned methods to segment the lungs, with the downstream goal of predicting chronic airflow limitation (CAL).
They try nn-Unet (with and without postprocessing), A Unet-based segmentor, TotalSegmentator and TriSwinUNETR. In order to predict CAL, they use the relative ratio of LAV950 (low hounsfield unit voxels < 950) in the different long lobes.

The results indicate that the different methods are relatively consistent in average volume of the different lobes. The CAL AUROC scores showed best results for the LL and LU lobes, with some slight variation across methods and lobes in general. The RM lobe which was associated with most anatomical variation, as well as being smaller than the other was worse at CAL detection.

**Strengths:**

- Interesting problem
- Good choice of models

**Weaknesses:**

- The text is generally quite hard to read, as it seems very unstructured.
- It would help the description of the different off-the-shelf models so much if it was a table instead. E.g. Type of data, what the model was trained to predict, #Samples trained on. And whether it was finetuned, or if the networks were adapted in other ways.
- Is LAV950 something that is predicted? E.g. from the abstract: "Results showed minimal differences in mean lung volume and LAV950 across methods", indicating that LAV950 varies across methods. Isn't LAV950 simply voxels with HU below 950 - I.e. computable with a threshold on the volume. If you mean that the ratio of LAV950 changes because the different lobes are segmented differently then it should be worded differently.
- It would be really nice to see a volume rendering showing LAV950 for e.g. a subject with CAL and without.
- Figure 1 needs a legend to specify which colors are which lobes.

---

### Official Review · Reviewer_CmSa · 2025-11-03

**Soundness:** 3
**Correctness:** 3
**Rating:** 1
**Confidence:** 4

**Summary:**

Applies five different segmentation methods on a lung volume prediction dataset.

**Strengths:**

The abstract is well-structured and well-written.

**Weaknesses:**

The abstract does not suggest any new work in ML, but rather applies five existing methods and presents the results without much technical details or discussion. The work might be more relevant in a computer science and medical joint conference, but it does not seem very relevant for a deep learning conference.

---

### Decision · Program_Chairs · 2025-11-05

**Decision:**

Accept

**Comment:**

The reviewers’ concerns carry less weight because the submission is an abstract and reflects ongoing or partial work. The PCs consider the work interesting and appropriate to share at the conference, as noted by the reviewer 3fmJ.